Subject Areas:
nanotechnology/quantum physics/electron microscopy

Keywords:
field emission, scanning tunnelling microscopy, current–voltage characteristics, surface imaging

Author for correspondence:
D. Pescia
e-mail: pescia@solid.phys.ethz.ch

†Present address: Istanbul Technical University, Department of Physics, Maslak 34469, Istanbul, Turkey.

# Non-topographic current contrast in scanning field emission microscopy

G. Bertolini[1], O. Gürlü[1,†], R. Pröbsting[1], D. Westholm[1], J. Wei[1], U. Ramsperger[1], D. A. Zanin[1], H. Cabrera[1], D. Pescia[1], J. P. Xanthakis[2], M. Schnedler[3] and R. E. Dunin-Borkowski[3]

[1]Laboratory for Solid State Physics, ETH Zurich, 8093 Zurich, Switzerland
[2]Electrical and Computer Engineering Department, National Technical University of Athens, Athens 15700, Greece
[3]Ernst Ruska-Centre for Microscopy and Spectroscopy with Electrons and Peter Grünberg Institute, Forschungszentrum Jülich, 52425 Jülich, Germany

 OG, 0000-0001-8831-1775; JW, 0000-0003-4146-0604; HC, 0000-0002-5590-1186; DP, 0000-0001-7436-418X; JPX, 0000-0002-8283-4011; RED-B, 0000-0001-8082-0647

In scanning field emission microscopy (SFEM), a tip (the source) is approached to few (or a few tens of) nanometres distance from a surface (the collector) and biased to field-emit electrons. In a previous study (Zanin et al. 2016 Proc. R. Soc. A **472**, 20160475. (doi:10.1098/rspa.2016.0475)), the field-emitted current was found to change by approximately 1% at a monatomic surface step (approx. 200 pm thick). Here we prepare surface domains of adjacent different materials that, in some instances, have a topographic contrast smaller than 15 pm. Nevertheless, we observe a contrast in the field-emitted current as high as 10%. This non-topographic collector material dependence is a yet unexplored degree of freedom calling for a new understanding of the quantum mechanical tunnelling barrier at the source site that takes into account the properties of the material at the collector site.

## 1. Introduction

By the physical process of field emission, electrons, confined within a metal, tunnel quantum mechanically through a classically forbidden potential barrier, surrounding the emission site, to escape into vacuum [1–3]. The barrier width is decreased by the application of a suitably strong electric field, so that the phenomenon is also referred to as electric field-assisted quantum tunnelling. The electron beam produced in this way is used e.g. in the column of

the most advanced electron microscopes [4]. If a 'collector' is placed a few nm to a few tens of nm distance from the emission site, a nanoscale junction is established that functions as the building block of a nanoscale vacuum channel transistor [5,6]. The proximity between tip and collector restricts the lateral size of the field-emitted beam: a lensless, 'nanoscale' electron microscope originates [7–9]. By recording the field emission current and the current of electrons escaping the tip–vacuum–collector junction, one atomic layer thick asperities at the collector site were imaged with 1 nm lateral spatial resolution [9]. Scanning tunnelling microscopy (STM) [10] (where the tip is held at subnanometre distances from the surface) achieves a better lateral spatial resolution but lacks the electronic system of the escaping electrons that appear in scanning *field emission* microscopy (SFEM) [7–9].

The quantitative understanding of field emission goes back to the dawn of quantum mechanics [1]. It foresees that the field-emitted current density is proportional to $e^{-G}$, the Gamov factor $G$ behaving approximately as $F^{-1}$, with $F$ being the electric field at the apex of the tip [1,2]. $F$ is proportional to the magnitude of the electrostatic potential difference between the source and the collector. The proportionality constant [1,2,11] $\beta$ depends primarily on the geometry of the source: for instance, it can be enhanced by sharpening the tip to nanometre scale radius of curvature [11]. Vertical asperities at the collector site change the effective distance $Z$ between tip and collector, thus affecting $\beta$ ([12,13] and references therein). The $Z$-dependence of $\beta$ has been verified experimentally [9]: a monatomic surface step (approx. 200 pm), imaged in SFEM at an average distance of 6–10 nm, changes the field-emitted current by approximately 1%.

In this work, we have prepared surfaces with different materials residing next to each other on nanometre lateral scale, with the aim of determining the limits of vertical resolution in SFEM. The dual system primarily studied here consists of $p(1 \times 1)$ tungsten domains (referred to for simplicity as 'W') versus $R(15 \times 12)$ tungsten carbide-domains ('WC'), appearing on a W(110) surface upon a suitable sample handling protocol. The vertical corrugation introduced by C-reconstructed domains is less than 15 pm. Surprisingly, when we move the tip from a W domain to a WC domain, we observe changes of the field-emitted current of approximately 10%, i.e. two orders of magnitude larger than expected on the base of $Z$-changes. Preliminary results have been published in a conference proceedings [14].

## 2. Experimental results

During the STM and SFEM experiments, the base pressure was less than $2.0 \times 10^{-11}$ mbar and the sample was at room temperature. The surface topography is detected, in this work, by STM imaging. STM is performed in the constant current mode, i.e. the tip is displaced vertically by a feedback loop in order to keep the tunnelling current constant ('red' in the schematic illustration figure 1a of the 'constant current' STM mode). The applied voltage $U$ is typically less than 1 V and the tunnelling currents are approximately 300 pA. The subsequent field emission imaging is primarily performed in a 'constant height' mode (figure 1b): the software interpolates the tip displacements, encoded during previous STM imaging, as a function of the lateral coordinate, by means of a mathematical plane. This defines a planar coordinate system parallel to the previously imaged area along which the tip is translated during field emission imaging. The quantity recorded in this mode is no longer the vertical tip translation but the current absorbed by the collector ($I_c$, red in figure 1b) or the current field-emitted by the source ($I_s$, blue in figure 1b).

The dual system on which we have focused the experimental work consists of domains of a two-dimensional tungsten-carbide reconstruction (technically: $W(110)/C - R(15 \times 12)$ [15,16], referred to here as 'WC' for simplicity) embedded within large regions of the 'clean', $p(1 \times 1)$ phase ('W') of a W(110) single crystal surface. The preparation of the dual W-WC sample starts with the cleaning of the W(110)-single crystal [15]. The bulk of a W crystal always contains a few per cent of C left behind during crystal growth. When the sample is flashed to approximately 2300 K, C segregates to the surface and builds locally two surface carbides known as $W/C - R(15 \times 3)$ and $W/C - R(15 \times 12)$ [15,16] reconstructions. After repeated flashing we find predominantly the $W/C - R(15 \times 12)$ reconstruction. If the cooling down is fast enough, the $W/C - R(15 \times 12)$ reconstruction, which is referred to as metastable in [15], is preserved and is found locally next to regions of 'clean', $p(1 \times 1) - W(110)$; see, for example, the STM image figure 2a. The formation of the $W/C - R(15 \times 12)$ nanomesh is revealed in the STM image figure 2b (not atomically resolved) by the typical pattern of periodically ordered larger and smaller bright spots, arranged along 'rows' repeating approximately every 3 nm. We observe that the average vertical corrugation in the carbidic phase is approximately 10 pm. The relative displacement of the two phases is approximately 5 pm (figure 2c).

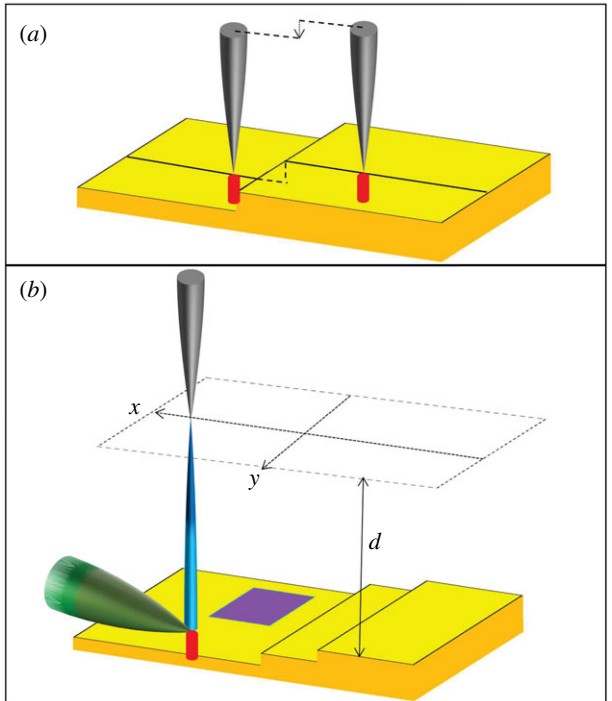

**Figure 1.** (*a*) Schematic view of the constant current STM imaging mode. The tip (grey) is displaced vertically (dashed arrow) when moved across a monatomic step on the surface (profiled by a dashed line) of the sample (yellow), so that the STM tunnelling current ('red' beam) is kept constant. (*b*) Schematic view of the constant height SFEM imaging. The tip is moved along a surface (dashed) which has a constant average distance $d$ from the underlying surface (consisting of monatomic steps and including a 'purple' domain). The 'blue' beam represents the electrons field-emitted from the tip ($I_s$). The 'red' beam represents the electrons entering the sample ($I_c$). The 'green' beam shows those electrons that escape the tip–surface junction.

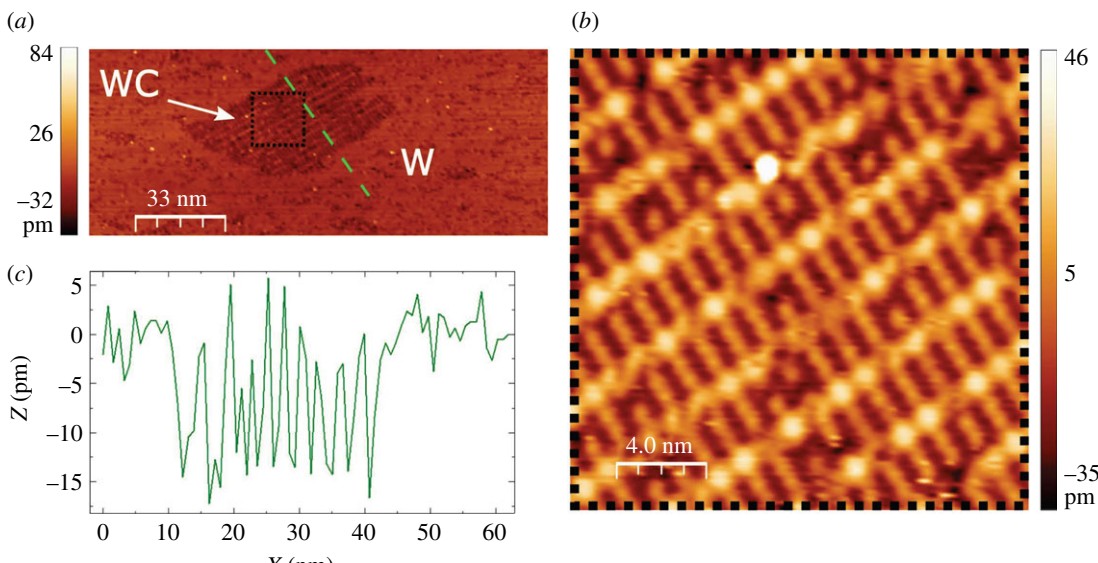

**Figure 2.** (*a*) STM topography of a W(110) surface. The tip vertical displacement is rendered with the colour code specified in the vertical bar. In the middle of the terrace, a domain of embedded carbide is visible. (*b*) 20 × 20 nm zoom of the black dotted frame in *a* showing the $R(15 \times 12)$ carbide reconstruction. (*c*) Height profile across the carbide domains (along the green path in *a*) revealing a maximum corrugation of less than 15 pm when moving from one region to the other. Scanning parameters: $U =$ 0.9 V, current: 500 pA.

The SFEM experiment starts with the harvesting of the WC domains by conventional STM. Figure 3*a* shows a set of W terraces separated by monatomic steps, running along the diagonal of the image. As STM imaging is performed in constant current mode, the colour code describes the relative height

**Figure 3.** Field emission imaging. (*a*) STM image of the W(110) surface showing terraces separated by monatomic steps. The colour code in the vertical bar gives the position of the terraces with respect to the lowest one (darkest) on the right. The carbidic phase appears as a faint corrugation on some terraces. (*b*) The same region of the surface is imaged in the field emission regime ($U =$ 41 V, $Z = 13$ nm). The colour code used to encode the absorbed current is given in the vertical bar. (*c*) The field emission image 3*b* (the current being divided by the line average) is superimposed onto the STM image 3*a* for clarity. The colour code used to encode the normalized current is given in the vertical bar.

of the terraces with respect to the lowest one on the right-hand side. There are some point defects spread across the image, but also domains with a faint, streak-like corrugation, hosting the carbidic reconstruction (see e.g. the middle terrace). The reconstruction appears 'faint', i.e. in the vertical scale used to render the monatomic steps (which are approx. 200 pm thick, i.e. approx. 10 times thicker than the WC corrugation). Figure 3*b* was taken over the same surface region depicted in figure 3*a* but in the field emission regime ($Z = 20$ nm, $U = 41$ V). The image displays $I_c$, coded as specified by the vertical bar. The regions corresponding to the WC domains appear brighter, indicating an enhancement of $I_c$. For this image, the average $I_c$ contrast, defined as $(I_c^{WC} - I_c^{W})/(I_c^{WC} + I_c^{W})$ between WC and W (the extraction of the contrast from images is explained in appendix A) is approximately 17%. The image is taken by scanning along horizontal lines, from the top to the bottom. During imaging (the time for scanning one horizontal line, consisting of 256 pixels, is typically 2 s) instabilities of the field-emitted current introduce noise into the image. To partially eliminate this noise, $I_c$ of figure 3*b* is divided by the average value along the horizontal line and plotted (colour code in the vertical bar) in figure 3*c* (superposed onto the STM image). As the images show, the change of $I_c$ is as local as the boundary between WC and W (approx. 5 nm, see appendix A).

For technical reasons, in figure 3 we did not measure $I_s$ but $I_c$. Therefore, a situation is possible where $I_s$ is the same over WC and W, and the contrast in $I_c$ is compensated by a contrast carried by those electrons that escape the junction [9]. Two experimental facts prove that this situation does not occur. First, we have imaged the same domains *simultaneously* to $I_c$ using those electrons that escape the junction ('green' in figure 1*b*, see also appendix B). We observe an enhanced count rate on top of WC in this channel of detection as well, producing a contrast with the same sign as $I_c$. Second, we have implemented the simultaneous recording of $I_c$ and $I_s$ in a different experiment and taken images of the same surface region at different values of $U$ in both channels of detection: $I_c$ and $I_s$. The two currents are necessarily measured with two different current amplifiers which are only nominally identical but, in practice, can have some instrumental multiplicative offset. We estimate this offset to be approximately 1%. The resulting $I_c(U)$ and $I_s(U)$ are displayed in figure 4*a*. The values of the

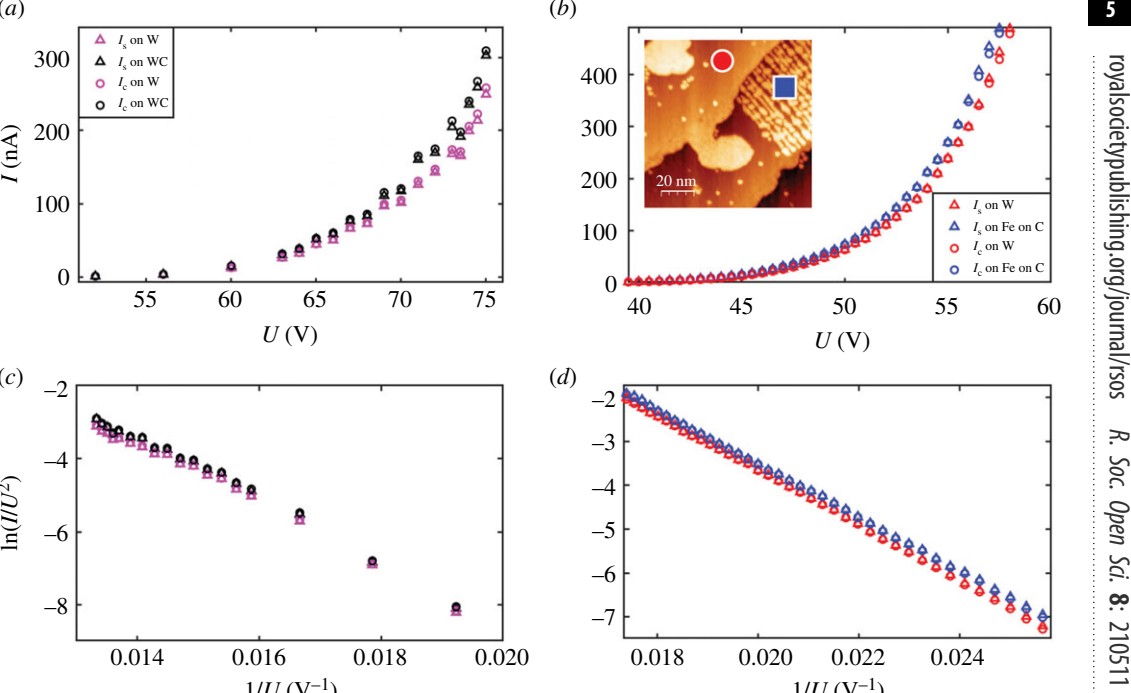

**Figure 4.** Current–voltage characteristics. (a) Obtained by spatial average of images of WC and W domains taken at $Z = 26$ nm. Black triangles: $I_s$ on WC. Black circles: $I_c$ on WC. Pink triangles: $I_s$ on W. Pink circles: $I_c$ on W. (b) $I_s$ and $I_c$ as a function $U$ for Fe on WC (blue triangles, respectively, blue circles) and W (red triangles, respectively, red circles), $Z = 30$ nm. Inset: STM image of the domains. The blue square (Fe on WC) and red circle (W) are the spots on top of which the current–voltage characteristics of figure 4b were measured. (c and d) The experimental data shown in a and b are plotted in a semilogarithmic scale (as Fowler–Nordheim plots) for more clarity in the low-voltage regime.

current were obtained by averaging over the respective domains. Within the margin of instrumental uncertainty, the currents in the two channels coincide and both are enhanced on WC. We can therefore state with certainty that the region of the *collector* residing just in front of the tip apex participates in the formation of the tunnelling barrier at the *source site*. The dependence on the collector of the field-emitted current with a similar order of magnitude is recorded systematically for further dual systems, see appendix D. As an example, we plot in figure 4b $I_s(U)$ and $I_c(U)$ curves recorded on a dual system consisting of W (red circle in the inset) versus Fe-covered WC (blue square). The sample preparation is explained in appendix D. In figure 4b, the current–voltage characteristics are taken by keeping the tip at a fixed position on top of the two different surface domains. The elimination of current fluctuations and noise is particularly important when comparing simultaneously measured $I_s(U)$ and $I_c(U)$ curves. Since the current is measured on the same wire that brings the bias to the tip or the sample, when one of the two electrodes is subject to a change of the bias in time the current measured on it is also affected by such change, showing possibly a capacitive response. Accordingly, the system needs a certain amount of time to relax after setting a certain voltage. In our system, this relaxation time is required on the $I_s$ channel when the tip voltage is varied. We estimated that a relaxation time of 100 ms between consecutive current measurements is needed after $U$ is changed. Using shorter times resulted in current–voltage characteristics measured forwards (i.e. by increasing $U$) and backwards (i.e. by decreasing $U$) not to coincide. The acquisition time was set, accordingly, to 100–200 ms. We learn from figure 4a,b that (i) $I_s$ and $I_c$ almost coincide, leaving the number of electrons escaping the junction in the 1% range, and (ii) the field-emitted current contrast is significantly larger than expected from the vertical corrugation.

## 3. Discussion

A 10% current contrast can be achieved if one assumes that some characteristics of the tip, such as its work function $\phi_{\text{tip}}$ (we recall that $G \propto (\phi_{\text{tip}})^{3/2}$ [1]) or its effective emission area (entering the current as a prefactor to the exponential) change when the tip is moved from one domain to the other. In fact, one can obtain a good fit of the experimental $I(U)$ curves of figure 4 by adapting the tip work function. However, during

imaging, such a change must occur reversibly and exactly in correspondence with the domain boundary. For example, a picking up of some atom by the tip when on top of WC should be followed by a release of it when the tip is on W and a picking up of the same atom in the same position again when re-entering the WC domain. A reversible change of the tip parameters therefore seems very improbable, also because one should find it for any of the dual systems reported in appendix D.

The only parameters remaining for explaining the contrast are $\beta$ and $U$, entering the electric field $F$ at the emission site, $F = \beta \cdot U$. It is known that $\beta = \beta(Z)$ [9,12,13], i.e. a vertical corrugation on the collector site is bound to change $Z$ and with it the electric field at the apex [2,12,13]. In the present work, we have taken care of harvesting specifically those WC domains that are embedded within the *same terrace as the neighbouring W domains* (figure 2a), so that the only vertical corrugation left when going from WC to W is that entailed by the WC corrugation or by a displacement on the entire WC domains with respect to the W domains. This corrugation is, however, only a few pm (figure 2c) and can only produce a contrast of the field-emitted current that is approximately two orders of magnitude smaller [9] than that observed here.

As originally predicted by Simmons [17], the work function difference $\phi_{\text{tip}} - \phi_{\text{collector}}$ between tip and collector changes the magnitude of the electrostatic potential difference between the tip and collector from the applied voltage $U$ (the difference between the Fermi levels of tip and collector) to $(U + ((\phi_{\text{tip}} - \phi_{\text{collector}})/e))$ ($e$: magnitude of the electron charge). It is therefore necessary to ascertain any work function changes between W and WC domains. In appendix C, we have determined work function changes using the Gundlach regime of STM [18–21]. We find that $(\phi_{\text{WC}} - \phi_{\text{W}})/e \approx -0.1$ V. To explore in a quantitative way the role of $\phi_{\text{collector}}$ on the current, we simulate current–voltage characteristics for a model system consisting of a metallic tip (modelled as a hyperboloid of revolution) with work function 4.5 eV at a variable distance $Z$ from a planar collector. For the simulation of the current–voltage characteristics in the field emission regime, we employed a software package [22,23] that first solves the Poisson equation as well as the drift and diffusion equations for a tip–vacuum–sample system in three dimensions by a finite-difference scheme. The sample surface is assumed to be planar, while the metal tip is modelled as a hyperboloid of revolution. Image potential terms are included. The resulting electrostatic potential along the central axis through the tip apex is used, in a second step, to derive a current through the vacuum barrier by employing the model of Bono and Good [24,25]. In their model, the transmission probability is approximated by ordinary Wentzel–Kramer–Brillouin (WKB) formulae. For electrons with a kinetic energy (in normal direction) above the vacuum barrier, the transmission probability is one. Accordingly, this model can be seen as a first approximation of the field emission current in the Fowler–Nordheim regime. A tip radius of 4 nm and opening angle of 7° were chosen, to better suit the experimental situation [13]. The work function of the tip was set to 4.5 eV. The work function of the sample was set to either 4.0 or 3.5 eV. The software package was developed primarily for semiconducting samples. In order to approximate a metal sample, we assumed a negligibly small band gap of 0.1 eV (negligible with respect to the work function change assumed in the simulations), a density of states effective mass of 1 and a carrier concentration of the order of $10^{20}$ cm$^{-3}$. Furthermore, in order to rule out a penetration of the electric field into the sample completely, the relative permittivity was set to a value close to zero. The inset of figure 5 shows simulated $I(U)$ for $Z = 13$ nm and for $\phi_{\text{collector}} = 4.0$ eV (squares) and $\phi_{\text{collector}} = 3.5$ eV (circles). The 'square' data points can be fitted by a Fowler–Nordheim functional dependence $I = a \cdot U^2 \cdot e^{-(b/U)}$ ($a$, $b$ being some suitable parameters). The circular data points can be fitted with the *same* functional law, with the *same* fitting parameters $a$, $b$ but with $U$ replaced by $U - 0.4987$ V. The potential shift of $-0.4987$ V reproduces the collector work function difference of $-0.5$ eV assumed for the simulations. The simulated data have a general comprehensive symmetry: all simulated data points, computed for $Z = 4$, 9, 11, 13, 20 nm and for $\phi_{\text{collector}} = 3.5$ eV and $\phi_{\text{collector}} = 4$ eV collapse, almost perfectly, onto one single graph when the current is plotted as a function of the shifted and rescaled voltage $\tilde{U} \doteq R(Z) \cdot (U + c)$ (figure 5). The multiplicative factor $R(Z)$ corrects for the different values that the parameter $b$ acquires at different distances [13]. The shift parameter $c$ is 0 for $\phi_{\text{collector}} = 4.0$ eV and $c \approx -0.5$ V for $\phi_{\text{collector}} = 3.5$ eV, independent of $(Z, U)$. The systematic *rigid* shift used to explain the simulated data points is in line with Simmons' analytic results [17]. Notice that for the simulations we have assumed a work function difference of 0.5 eV for the convenience of display: in fact, in the simulations, the experimental work function difference of 0.1 eV would produce barely distinguishable $I(U)$ curves! We conclude that the experimentally observed current contrast is too large to be explained by a difference of the work function on the collector site, although, taking into account all data, also those shown in appendix D, it seems that the sign of the current contrast is determined by the sign of the difference between the work function, namely, the source current

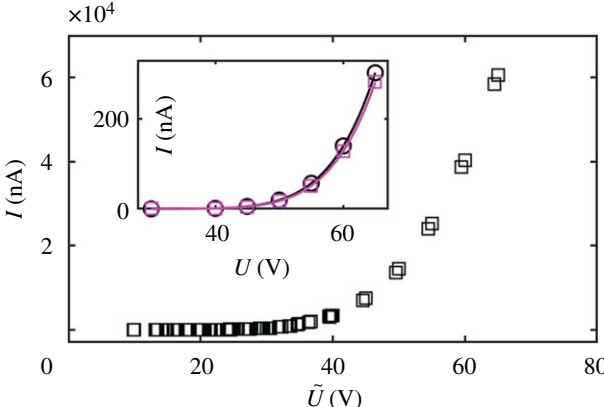

**Figure 5.** Simulated current–voltage characteristics. Inset: $Z = 13$ nm, circles: $\phi_{collector} = 3.5$ eV, squares: $\phi_{collector} = 4.0$ eV. Continuous lines are fits with a Fowler–Nordheim functional dependence $I = a \cdot U^2 \cdot \exp(-b/U)$ (squares) and $I = a \cdot U^2 \cdot \exp(-(b/(U+c)))$ (circles). Bulk: All simulated data points ($Z = 4, 9, 11, 13, 20$ nm and $\phi_{collector} = 3.5, 4.0$ eV) plotted as a function of the rescaled variable $\tilde{U} = R(Z) \cdot (U - c)$. $c = 0$ for $\phi_{collector} = 4.0$ eV, $c = -0.5$ V for $\phi_{collector} = 3.5$ eV.

increases with a decrease in the work function of the collector. We are therefore left with a result which we do not completely understand, at least from the quantitative point of view.

There is, however, a degree of freedom that we have not considered yet. We know that the electric field at the source site is approximately 6 V nm$^{-1}$ during field emission [13]. By virtue of the boundary conditions, an electric field at the tip translates to the region of the collector residing in front of the tip. The electric field at the collector site, which is only a few nanometres away from the tip, is still sizeable (approx. 2 V nm$^{-1}$ [26]). An electric field at the surface of a metal, pointing from the metal to the vacuum, is known to increase the work function approximately linearly with its strength. The change is material dependent [27–32] and can be of the order of 0.4–0.8 eV @ 2 V nm$^{-1}$ (see e.g. fig. 3 in [27]). This new degree of freedom introduces the possibility that the applied potential $U$ is not rigidly shifted as in the model of Simmons [17] but is shifted by a term that is linear in $U$, i.e. that the parameter $c$, assumed to be a constant in this work, is actually proportional to $U$. This mechanism introduces an effective dependence of $\beta$ (i.e. the fitting parameter $b$) from the material residing at the collector site. If one uses this new degree of freedom to fit the measured $I(U)$ characteristics, one finds that a relative change $(\beta_{WC} - \beta_W)/\beta_W$ of the order of approximately 3–8% explains the observed current–voltage characteristics better than a rigid shift. However, we do not have any sign of a material-dependent work function change in the low and intermediate voltages modes of STM. A more quantitative discussion that treats on an equal footing the various regimes of STM (the tunnelling, the Gundlach and the field emission regime) is therefore, at this point, mandatory.

Data accessibility. The ad hoc software used to analyse the images is available at Zenodo: https://zenodo.org/record/3675444#.YOMsNn7TWUk [33]. All data required to support the conclusions of the presented research are contained in the included figures.

Authors' contributions. G.B., O.G. and R.P. did the experiments reported in figures 2, 3, 4b, 4d, 6, 7, 8, 9, 10 and 12. D.W, J.W. and U.R. did the experiments reported in figures 4a, 4c and 11. D.A.Z. and U.R. built the instruments used in this work. H.C. discussed the paper. M.S. did the simulations reported in figure 5. M.S., J.P.X. and R.E.D.-B. provided the theoretical input and discussed the paper. D.P. wrote the paper.

Competing interests. We declare we have no competing interests.

Funding. The research was partially funded by Marie Curie Initial Training Network (ITN), grant no. 606988 under FP7-PEOPLE-2013-ITN, the Swiss National Science Foundation (SNF grant no. 20-134422) and the Commission for Technology and Innovation (CTI grant no. 9860.1 PFNM-NM).

Acknowledgements. O.G. thanks M. Erbudak and E. Tosatti. D.P. thanks A. Kyritsakis and C. Walker for helpful discussions. D.P., G.B., J.W., D.W. and U.R. thank R. Forbes for helpful discussions.

# Appendix A. Width of the boundary and image contrast

For determining the width of the boundary between W- and WC-surface domains, as seen in field emission STM imaging, we have developed an ad hoc software [33]. We mark first, by visual inspection, the points that divide the W from the WC domains in the STM image figure 6a (blue line). For border detection in field emission images, we fit a function $A \cdot \tanh(B \cdot x + C) + D$ along each scanning line in figure 6b,c. The inflection points of the *tanh*-functions are given as green points in

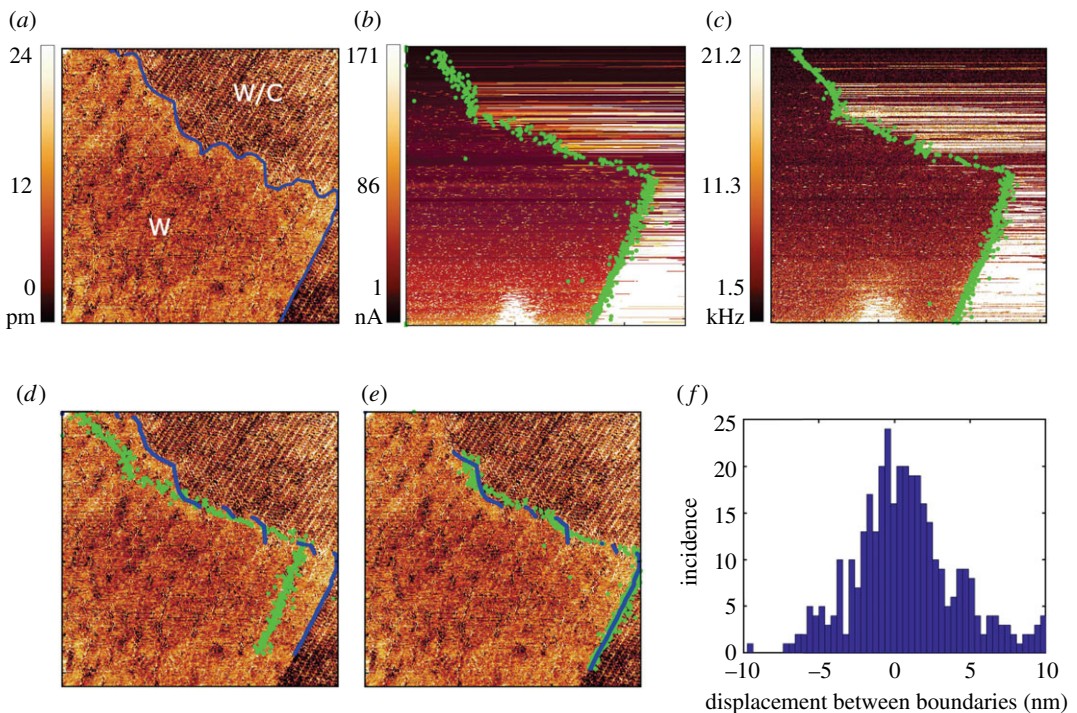

**Figure 6.** (*a*) STM image of the clean W surface (left) and the W(110)/$C - R$(15 × 12) phase (right). The images contain 512 × 512 px, which corresponds to 147 × 147 nm. The border between the two phases is marked with a blue line, by visual inspection. (*b*) The collector current image of the same region as in *a* ($U = 35$ V, $Z = 9$ nm). The border (green points) between W region (left) and carbide region (right) zones is determined by fitting a tanh-function along each line (see text). (*c*) Secondary electron maps, taken simultaneously with *b*. The green border is determined as in *b*. (*d*) The green border determined in *b* is inserted into the STM image. This image shows that, upon retraction of the tip from the tunnel regime to the field emission regime, the piezo-scanners drifted slightly. (*e*) By shifting and rotating the image, the two borders are made to overlap. (*f*) Distribution of the displacement between blue and green points. The full width at half maximum is approximately 5 nm.

figure 6*b,c* (for some scan lines the fit did not converge properly because some lines did not show any step). The width of the boundary is then estimated by comparing the boundaries determined in STM (which are assumed to be almost perfectly sharp in view of the intrinsic higher lateral spatial resolution of STM) with those obtained in field emission imaging. For proper comparison, the 'blue' line and the 'green' line (slightly mismatched, see figure 6*d*) must be rematched by rotating and shifting the field emission images. Visual inspection of the data showed that offsets along the horizontal and vertical axes by 25 nm (respectively, 5 nm) and a clockwise rotation by 4.5° are suitable (figure 6*e*). The width of the boundary is then obtained by plotting the distribution of displacements between the blue and green boundaries (figure 6*f*) and determining the full width at half maximum, which turns out to be 5 nm. This is the width of the boundary as observed in field emission imaging. The width of the boundary is both an upper limit for the spatial range over which the potential shift occurs and the spatial resolution of the present field emission-STM experiment. In a previous study [9], the spatial resolution of field emission STM was determined to be approximately 1 nm, at best.

We define the average contrast between two average signals $Q_A$ and $Q_B$ acquired on two domains $A$ and $B$ by $(Q_A - Q_B)/(Q_A + Q_B)$. Once the boundary between the phases is determined, the average contrast is obtained by computing the total signals $Q_A$ and $Q_B$ on the two phases. The average contrast obtained by sampling 19 images recorded in the past 2 years for $A =$ WC and $B =$ W is approximately 10%, the lowest value measured being 3% and the highest value approximately 30%.

## Appendix B. Field emission current and secondary electron current in the dual system W−WC

The detection of the secondary electrons escaping the tip-target junction allows the taking of secondary electron images simultaneously to recording the absorbed current [9]. Figure 7 is an example of

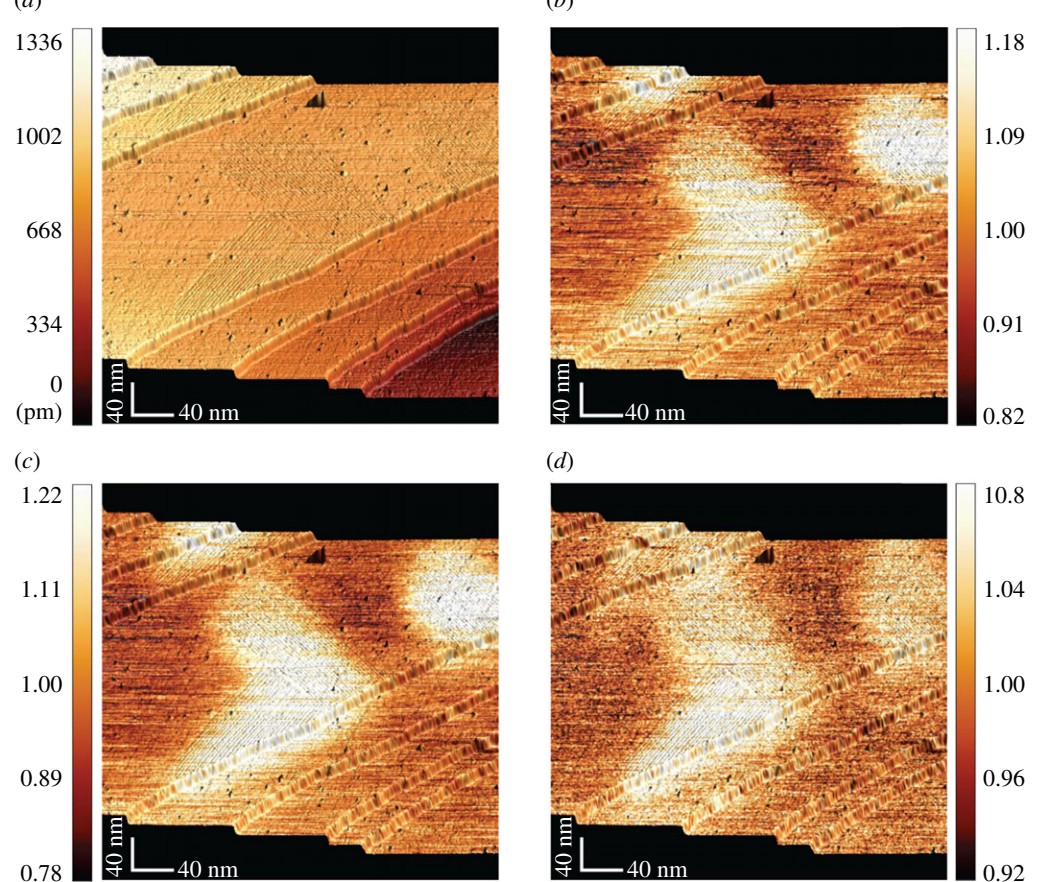

**Figure 7.** (*a*) STM image of W(110) terraces with a faint contrast due to $W(110)/C - R(15 \times 12)$. (*b*) The corresponding collector current image taken in the field emission regime, superposed onto the STM image ($U = 41$ V, $Z = 20$ nm). (*c*) The secondary electron image, superposed on the STM image *a* ($U = 41$ V, $Z = 20$ nm). (*d*) The secondary electron count rate of *c* has been divided by the absorbed current of *b*. In *b,c,d*, the signal is divided by the line average and the colour code used is specified in the corresponding vertical bar.

such comprehensive imaging. Figure 7*a* is the STM topography of several terraces of W (see also figure 3*a*), with some carbidic ($WC - R(15 \times 12)$) domains visible as a faint corrugation in the middle terrace. Figure 7*b* is the collector current image superposed onto the STM topography (see also figure 3*c*). Figure 7*c* is the secondary electron image, also superposed onto the STM topography. The count rate at each pixel has been divided by the line average, to eliminate some noise. Carbidic domains are brighter than W domains. In figure 7*d*, the secondary electron count rate has been divided by the collector current. Some contrast is still evident, indicating that the cross section for production of secondary electrons depends on the target material involved in the scattering process [9].

# Appendix C. Work function change in the W−WC dual system by means of Gundlach oscillations

The work function change ($\phi_{WC} - \phi_W$) between the two domains is determined from the energy shift of the higher order Gundlach oscillations [18–21], following the experimental method established in [19]. This method entails measuring the tip displacement $Z$, required to keep a preset tunnelling current constant, as a function of the applied voltage (figure 8*a*). Figure 8*b* shows the derivative of the $Z(U)$ curves taken on W and WC, obtained by a numerical algorithm. The shift of the peaks is shown in the inset as a function of the peak number. It is observed to level at higher order ($i > 2$) to a value of approximately $-0.1 \pm 0.02$ V, which is, accordingly, taken as a measure of ($\phi_{WC} - \phi_W$).

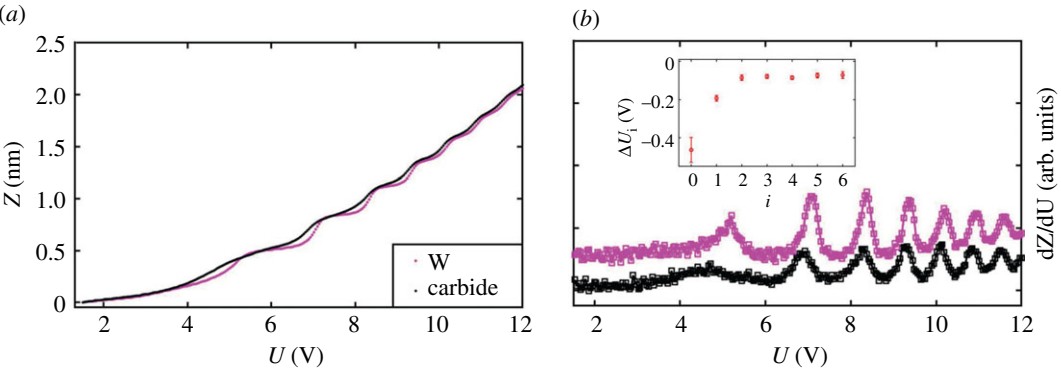

**Figure 8.** (*a*) $Z(U)$ curves taken with the tip residing on top of W and WC. (*b*) Numerical derivative of the $Z(U)$ curves showing seven Gundlach oscillations (numbered $i = 0$ to $i = 6$ from left). Pink squares: taken on W. Black squares: taken on WC. The inset shows the voltage shift as a function of the index '$i$'.

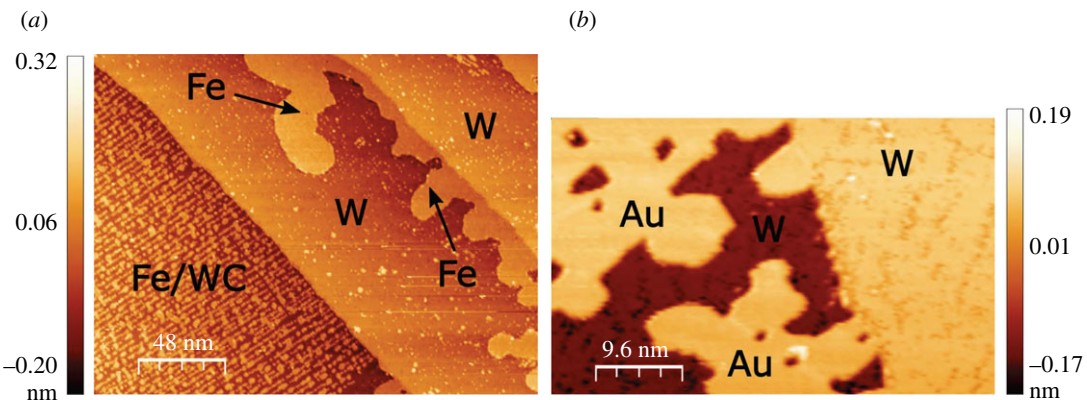

**Figure 9.** (*a*) STM image of 0.2 monolayers of Fe grown on W(110). Three domains are visible: clean W, Fe on clean W(110) (marked by arrows) and Fe on $W(110)/C - R(15 \times 12)$. The tip displacement is encoded with a colour specified in the vertical bar. (*b*) STM image of 0.2 Au monolayers grown on W(110). The right-hand side is a clean W(110) terrace. The left-hand side, at the same vertical level as the W(110) terrace, is the Au overlayer. The darker region in between is the lower lying W(110) substrate.

# Appendix D. Experimental results on further dual systems

Fe and Au are deposited on top of clean W(110) and on carbidic sections by molecular beam epitaxy at evaporation rates of approximately 0.20 monolayers $min^{-1}$. The sample during deposition has a temperature estimated to be between 470 and 570 K. The growth of submonolayer films of Fe on tungsten at room temperature produces one-monolayer-thick Fe islands. At higher substrate temperatures the Fe atoms diffuse, on clean W(110), to form larger one-monolayer-thick islands, mostly decorating the W surface steps (figure 9*a*, 0.2 monolayer (ML) Fe). On carbidic domains, the growth of Fe is affected by the raw-like corrugation (figure 9*a*). There, the Fe atoms aggregate to nanoclusters, which form chains aligned along high symmetry directions of the carbidic superstructure. The growth of Au on tungsten is approximately layer by layer as in the case of Fe. At higher substrate temperatures, Au atoms diffuse and aggregate in one-atom-thick layers, starting from the step edges of W(110) (figure 9*b*).

Figure 10 presents results of imaging submonolayer-thick Fe overlayers on W and WC. Figure 10*a* is the STM topography. Figure 10*b* is the collector current (divided by the line average). The Fe-covered domains are brighter that the uncovered W domains, the contrast being in the 10% range. As in the dual system W–WC, we observe a sizeable contrast in the collector current mode. Figure 10*c* is the secondary electron image taken simultaneously to figure 10*b*. The range of values used in the vertical bar of figure 10*c* is larger than the range used to represent the collector current. Accordingly, in figure 10*d* (where the secondary electron count rate is divided by the collector current), a contrast is still evident, indicating that also in this dual system the cross section for the production of secondary electrons is material dependent [9].

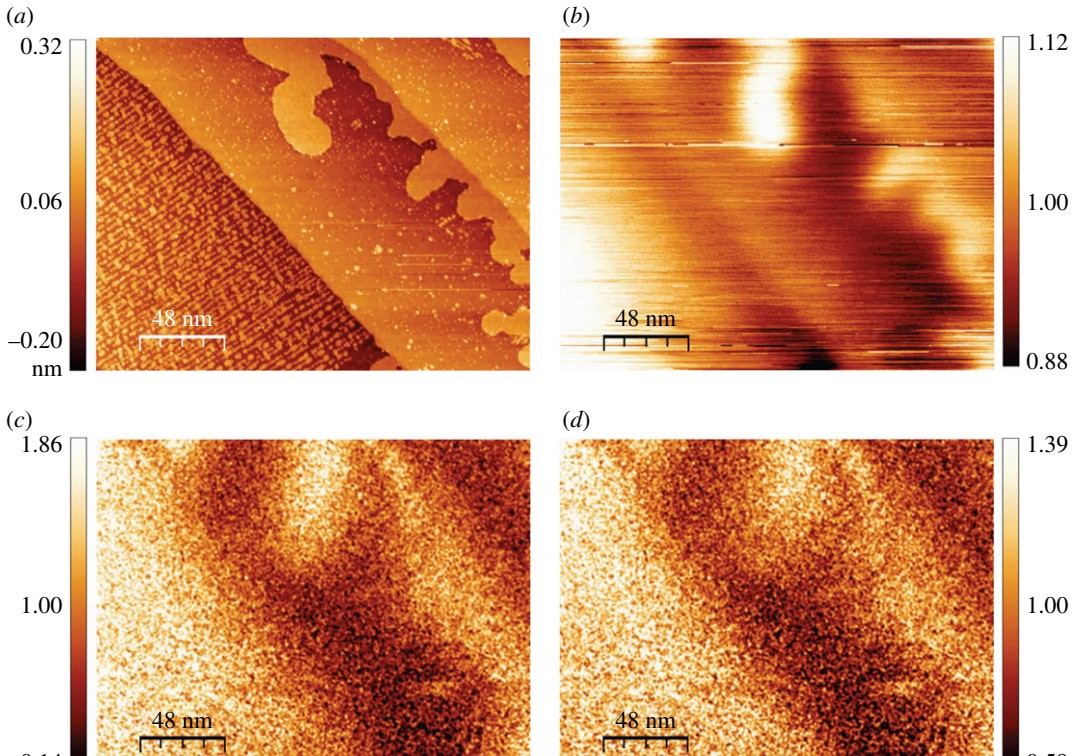

**Figure 10.** (*a*) STM image of W(110) terraces with Fe overlayers. The vertical bar gives the colour code used to encode the tip displacement. The nanodot-like structure visible on the left-hand-side terrace consists of Fe dots decorating the $W(110)/C − R(15 \times 12)$ phase. Along the middle terrace the one-monolayer-thick Fe deposits are seen to decorate the step edge between the middle terrace and the large terrace on the right-hand side. (*b*) The corresponding collector current image (divided by line average, $U =$ 64 V, $Z = 9$ nm, average current: 126.6 nA), taken in the field emission regime (the colour code used to encode the absorbed current is given by the vertical bar). The sectors with Fe deposits are brighter. (*c*) The secondary electron image, displaying the count rate divided by the line average (average count rate: 3.59 kHz). The Fe deposits are brighter, the range of brightness, specified by the vertical bar, being larger that the range observed in *b*. (*d*) The secondary electron count rate of *c* has been divided by the collector current of *b*. The colour code used to encode the signal is specified by the vertical bar.

Figure 11 presents images of a further dual system, consisting of 8 ML Fe deposited on W(110). Figure 11*a* is an STM image, showing the different morphology of the Fe film on WC ('granular', large terrace in the middle), and on W (elongated islands, terraces right and left of the middle one). Figure 11*b* is an STM image of the region of the sample that has been scanned successively in the field emission regime. Figure 11*c* is the secondary electron image of the region displayed in figure 11*b*. Brighter domains of enhanced secondary electron yield are in correspondence with the 8 ML Fe grown on WC. Figure 11*d* is the current image taken simultaneously with figure 11*c*. The brighter regions of enhanced collector current are in correspondence with the 8 ML Fe grown on WC.

Figure 12*a* shows a d$Z(U)$/d$U$-graph containing Gundlach oscillations, measured with the tip being kept on top of W (pink), Fe coating clean W (green) and Fe coating WC (dark green). From the energy shift of the four highest order oscillations we obtain $\phi_{Fe/WC} − \phi_W = −0.22 \pm 0.02$ eV and $\phi_{Fe} − \phi_W = −0.42 \pm 0.05$ eV. In figure 12*b*, Gundlach oscillations are reported on the dual system W and Au-coated W. From the energy shift of all peaks we obtain $\phi_{Au} − \phi_W = +0.15 \pm 0.05$ eV.

Figure 12*c* shows (right) $I(U)$ curves recorded on top of W (pink, the location of the tip during the taking of the data being marked by a dot in the image left) and Fe/WC (green, the location of the tip during the taking of the data being marked by a dot in the image left). The data points refer to different $Z$ (see caption). The $I(U)$ curves on Fe/WC are predominantly shifted by a negative $c$, in agreement with the brighter appearance of Fe/WC in the image figure 10*b*. We recall that $\phi_{Fe/WC} − \phi_W$ also has a negative sign, within experimental uncertainty. Figure 12*d* shows (right) $I(U)$ curves recorded on top of W (pink, the location of the tip during the taking of the data being marked by a dot in the image left) and Au (blue, the location of the tip during the taking of the data being marked

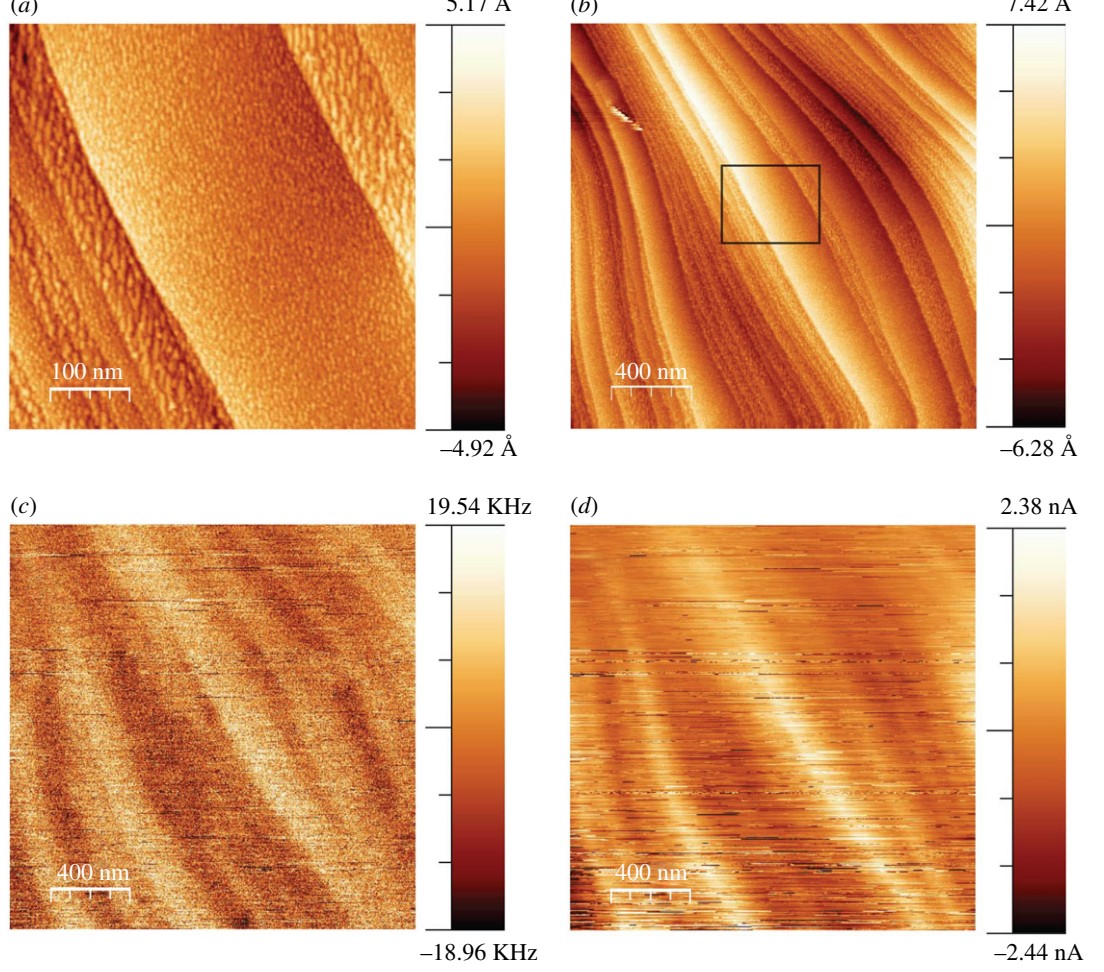

**Figure 11.** (*a*) STM image of W(110)-terraces with an 8 ML thick Fe overlayer. (*b*) STM image of the region of the W(110) surface scanned successively in the field emission regime. The section of the image zoomed in *a* is approximately indicated by a square framework. The vertical bars indicate the colour code used to encode the tip displacement in STM. (*c*) Secondary electron image of the surface region displayed in *b*. The vertical bar gives the colour code used to encode the change in secondary electron intensity ($U = 46$ V, $Z = 20$ nm, average current: 50 nA). (*d*) Collector current image taken simultaneously to *c*. The vertical bar gives the colour code used to encode the change of the field-emitted current.

by a dot in the image on the left-hand side). The data points refer to different $Z$ (see caption). In these data, the contrast appears to increase with $Z$. However, we do not attach a significance to this apparent increase, as the uncertainty of the data is quite large. A further set of experiments which studies systematically the $Z$ dependence is required to confirm or disprove the observed trend. What seems to be well established is that the $I(U)$ curves on Au are predominantly shifted by a *positive c*. We recall that, within experimental error, $\phi_W - \phi_{Au}$ has also a positive sign.

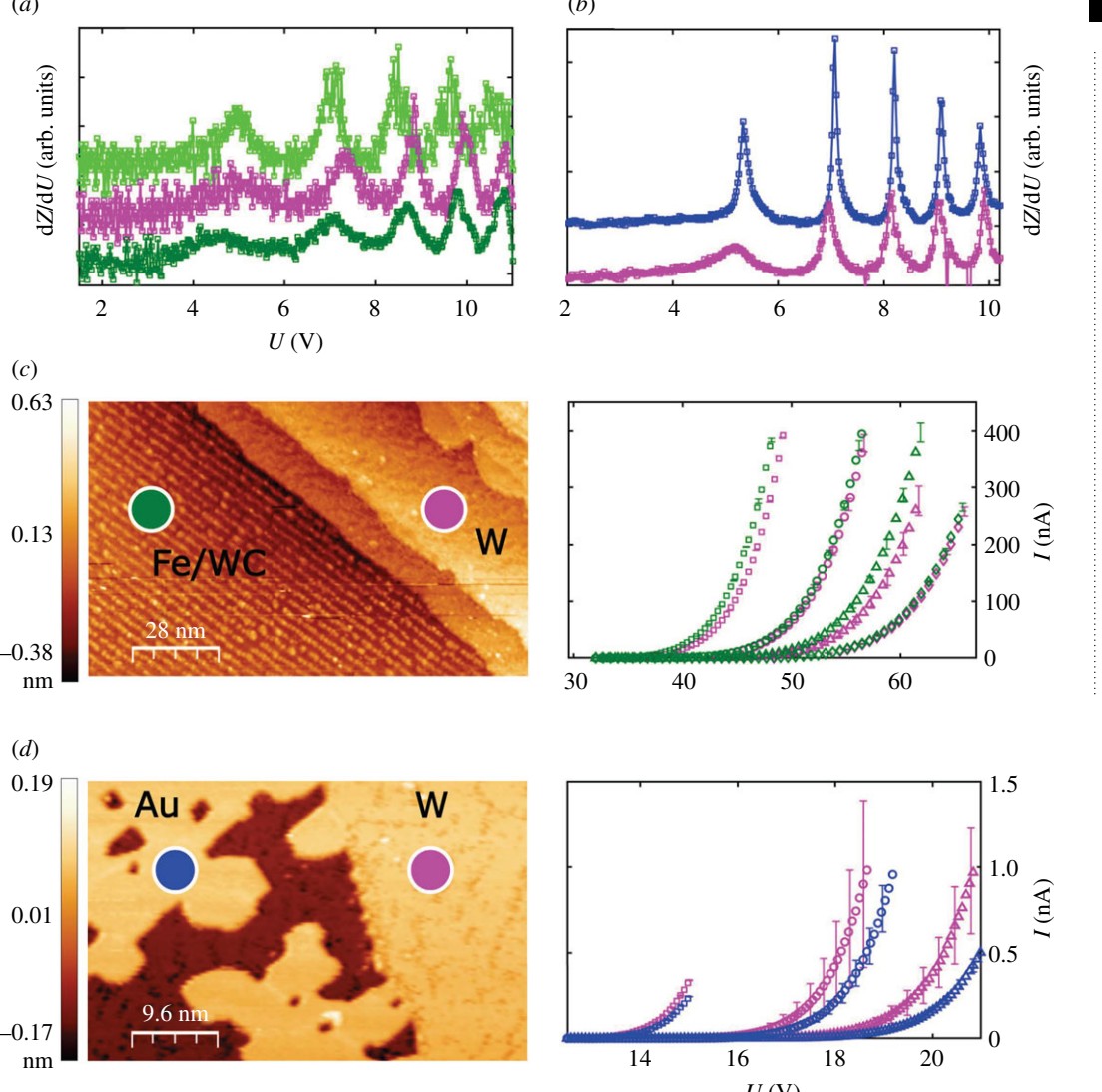

**Figure 12.** (*a*) Gundlach oscillations. Pink: on W. Green: on one monolayer thick Fe-deposits covering a clean W(110) terrace. Dark green: tip held on a Fe deposit covering $W(110)/C − R(15 \times 12)$. (*b*) Gundlach oscillations. Pink: on W. Blue: on an Au (one monolayer thick) deposit covering a clean W(110) terrace. (*c*) Left-hand side: STM image of one ML Fe on W. The current–voltage characteristics (right-hand side) were taken at the locations marked by dots (pink, W; Green, Fe covering the $W(110)/C − R(15 \times 12)$ phase). Squares: $Z = 11$ nm. Circles: $Z = 17$ nm. Triangles: $Z = 24$ nm. Diamonds: $Z = 30$ nm. (*d*) Left-hand side: STM image of one ML of Au on W. The current–voltage characteristics (right-hand side) were taken at the locations marked by dots (pink, W; blue, Au). Squares: $Z = 5$ nm. Circles: $Z = 8$ nm. Triangles: $Z = 11$ nm.

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
