## [Peer Review File · Royal Society Open Science]

Review History

RSOS-210511.R0 (Original submission)

Review form: Reviewer 1

Is the manuscript scientifically sound in its present form?

Yes

Are the interpretations and conclusions justified by the results?

Yes

Is the language acceptable?

Yes

Do you have any ethical concerns with this paper?

No

Have you any concerns about statistical analyses in this paper?

No

Recommendation?

Accept with minor revision (please list in comments)

Comments to the Author(s)

Attached (see Appendix A).

Review form: Reviewer 2

Is the manuscript scientifically sound in its present form?

Yes

Are the interpretations and conclusions justified by the results?

Yes

Is the language acceptable?

Yes

Do you have any ethical concerns with this paper?

No

Have you any concerns about statistical analyses in this paper?

No

Recommendation?

Accept with minor revision (please list in comments)

Comments to the Author(s)

The authors report an interesting experimental observation that the field emission current from the STM tip strongly depends on the properties of the scanned surface, even in the case of an almost perfectly flat topography of the sample. The authors claim that when the distance between the tip and the surface is several tens of nanometers, the potential barrier at the tip depends on the collector material to a much greater extent than the standard theory predicts for conventional metal-insulator-metal tunnel junctions. The presented results are of great interest to the vacuum electronics and STM communities.

A few minor comments:

- 1) Previous work of the authors should be mentioned: D. Westholm et al. "Non-topographic contrast in constant-current Scanning Field-Emission Microscopy", 2020, DOI: 10.1109/IVNC49440.2020.9203360
- 2) The pressure level and temperature of the experiments should be indicated.
- 3) The authors suggest that a high field of 2V/nm at the collector changes its work function. It would be worth giving a numerical estimate of the possible magnitude of such a change, for example, based on data from references [26-31].
- 4) It is not clear from the presented data whether there is a dependence of the current contrast on the applied voltage. It would also be more informative to plot some of the I(U) curves in semi-logarithmic coordinates so that the difference in current at low voltages is visible.
- 5) In Figure 11d, the current contrast appears to increase with Z. However, theory predicts the opposite relationship. It is worth commenting on this discrepancy.
- 6) It follows from Appendix D that the sign of the current contrast is determined by the sign of the difference between the work function, namely, the source current increases with a decrease in

the work function of the collector. If this statement is true, then it should be noted in the main text.

7) I believe there is a mistake in Page 8 line 51 “we recall that $\phi_W - \phi_{Fe}/WC$ also has a negative sign”, however it has positive sign according to Page 8 line 43 “ $\phi_{Fe}/WC - \phi_W = -0.22eV$ ”. The same for “ $\phi_W - \phi_{Au}$ ”.

Decision letter (RSOS-210511.R0)

Dear Professor Pescia

On behalf of the Editors, we are pleased to inform you that your Manuscript RSOS-210511 "Non-topographic current contrast in scanning field emission microscopy" has been accepted for publication in Royal Society Open Science subject to minor revision in accordance with the referees' reports. Please find the referees' comments along with any feedback from the Editors below my signature.

Please submit your revised manuscript and required files (see below) no later than 7 days from today's (ie 09-Jun-2021) date. Note: the ScholarOne system will 'lock' if submission of the revision is attempted 7 or more days after the deadline. If you do not think you will be able to meet this deadline please contact the editorial office immediately.

on behalf of Dr Peter Munro (Associate Editor) and Miles Padgett (Subject Editor)
openscience@royalsociety.org

Associate Editor Comments to Author (Dr Peter Munro):

Comments to the Author:

Thanks for your submission, please make the minor revisions suggested by the reviewers.

Reviewer comments to Author:

Reviewer: 1

Comments to the Author(s)

Attached. (Comments to the authors.pdf)

Reviewer: 2

Comments to the Author(s)

The authors report an interesting experimental observation that the field emission current from the STM tip strongly depends on the properties of the scanned surface, even in the case of an almost perfectly flat topography of the sample. The authors claim that when the distance between the tip and the surface is several tens of nanometers, the potential barrier at the tip depends on the collector material to a much greater extent than the standard theory predicts for conventional metal-insulator-metal tunnel junctions. The presented results are of great interest to the vacuum electronics and STM communities.

A few minor comments:

- 1) Previous work of the authors should be mentioned: D. Westholm et al. "Non-topographic contrast in constant-current Scanning Field-Emission Microscopy", 2020, DOI: 10.1109/IVNC49440.2020.9203360
- 2) The pressure level and temperature of the experiments should be indicated.
- 3) The authors suggest that a high field of 2V/nm at the collector changes its work function. It would be worth giving a numerical estimate of the possible magnitude of such a change, for example, based on data from references [26–31].
- 4) It is not clear from the presented data whether there is a dependence of the current contrast on the applied voltage. It would also be more informative to plot some of the I(U) curves in semi-logarithmic coordinates so that the difference in current at low voltages is visible.
- 5) In Figure 11d, the current contrast appears to increase with Z. However, theory predicts the opposite relationship. It is worth commenting on this discrepancy.
- 6) It follows from Appendix D that the sign of the current contrast is determined by the sign of the difference between the work function, namely, the source current increases with a decrease in the work function of the collector. If this statement is true, then it should be noted in the main text.
- 7) I believe there is a mistake in Page 8 line 51 "we recall that $\phi_{i_W} - \phi_{i_Fe/WC}$ also has a negative sign", however it has positive sign according to Page 8 line 43 " $\phi_{i_Fe/WC} - \phi_{i_W} = -0.22\text{eV}$ ". The same for " $\phi_{i_W} - \phi_{i_Au}$ ".

===PREPARING YOUR MANUSCRIPT===

===PREPARING YOUR REVISION IN SCHOLARONE===

-- Ensure that your data access statement meets the requirements at <https://royalsociety.org/journals/authors/author-guidelines/#data>. You should ensure that you cite the dataset in your reference list. If you have deposited data etc in the Dryad repository, please only include the 'For publication' link at this stage. You should remove the 'For review' link.

-- If you have uploaded ESM files, please ensure you follow the guidance at <https://royalsociety.org/journals/authors/author-guidelines/#supplementary-material> to include a suitable title and informative caption. An example of appropriate titling and captioning may be found at https://figshare.com/articles/Table_S2_from_Is_there_a_trade-off_between_peak_performance_and_performance_breadth_across_temperatures_for_aerobic_sc_ope_in_teleost_fishes_/3843624.

Author's Response to Decision Letter for (RSOS-210511.R0)

See Appendix B.

Decision letter (RSOS-210511.R1)

Dear Professor Pescia,

I am pleased to inform you that your manuscript entitled "Non-topographic current contrast in scanning field emission microscopy" is now accepted for publication in Royal Society Open Science.

on behalf of Dr Peter Munro (Associate Editor) and Miles Padgett (Subject Editor)
openscience@royalsociety.org

Appendix A

In this work, the authors presented a very interesting study of lateral field emission from tungsten-carbide surfaces. I recommend the publication in the Royal Society Open Science journal after minor revisions.

1. For the reader it is very difficult to visualize the measurements setup. The work would be more understandable if the authors add a Figure with a schematic, i.e. a tip at a certain distance from the surface. For instance, you can see an example of what I mean in this work: <https://doi.org/10.1002/aelm.202000838>
2. In the same way, a schematic of how the STM images were performed (constant current and constant height mode) can help the reader to better understand what I_c and I_s are.
3. How did you estimate the relaxation time of 100ms when varying the voltage on the tip?
4. Please carefully check the grammar and correct the typos. Moreover, some verb tenses are wrong.

Appendix B

Dear Editor,

we would like to resubmit the paper “Non-topographic current contrast in scanning field emission microscopy” by G. Bertolini et al to “Royal Society Open Science”.

Please find enclosed below our point-by-point reply to the comments of the referees (our reply is in bold letters). We thank them for their suggestions and comments.

Cordially

D. Pescia

REPLY TO REFEREE 1.

1. For the reader it is very difficult to visualize the measurements setup. The work would be more understandable if the authors add a Figure with a schematic, i.e. a tip at a certain distance from the surface. For instance, you can see an example of what I mean in this work:

<https://doi.org/10.1002/aelm.202000838>.

2. In the same way, a schematic of how the STM images were performed (constant current and constant height mode) can help the reader to better understand what I_c and I_s are.

Our reply to 1 and 2: we have included the required figure, now as Fig.1a,b, the corresponding caption and the following sentence at the opening of the section on experimental results.

“ During the STM and SFEM experiments, the base pressure was less than $2.0 \cdot 10^{-11}$ mbar and the sample was at room temperature. The surface topography is detected, in this work, by STM imaging. STM is performed in the constant current mode, i.e. the tip is displaced vertically by a feed-back loop in order to keep the tunneling current constant ("red" in the schematic illustration Fig.1a of the "constant current" STM mode). The applied voltage U is typically less than 1 V and the tunneling currents are about 300 pA. The subsequent field emission imaging is primarily performed in a "constant height" mode (see Fig.1b): the software interpolates the tip displacements, encoded during previous STM-imaging, as a function of the lateral coordinate, by means of a mathematical plane. This defines a planar coordinate system parallel to the previously imaged area along which the tip is translated during field emission imaging. The quantity recorded in this mode is no longer the vertical tip translation but the current absorbed by the collector (I_c , red in Fig.1b) or the current field emitted by the source (I_s , blue in Fig.1b).”

3. How did you estimate the relaxation time of 100ms when varying the voltage on the tip?

Our reply. We have added the following sentence: “ Using shorter times resulted in current voltage characteristics measured forwards (i.e. by increasing U) and backwards (i.e. by decreasing U) not to coincide.”

4. Please carefully check the grammar and correct the typos. Moreover, some verb tenses are wrong.

Our reply: we have done it as good as we could. A native UK citizen has provided help.

REPLY TO REFEREE 2:

1) Previous work of the authors should be mentioned: D. Westholm et al. “Non-topographic contrast in constant-current Scanning Field-Emission Microscopy”, 2020, DOI: 10.1109/IVNC49440.2020.9203360

Our reply: we have included the required reference and added the sentence “Preliminary results were published in a Conference Proceedings”

2) The pressure level and temperature of the experiments should be indicated.

Our reply: we have provided the required informations (see above).

3) The authors suggest that a high field of 2V/nm at the collector changes its work function. It would be worth giving a numerical estimate of the possible magnitude of such a change, for example, based on data from references [26–31].

Our reply. We have added the following sentence, as required by the referee: An electric field at the surface of a metal, pointing from the metal to the vacuum, is known to increase the work function approximately linearly with its strength. The change is material dependent\cite{Zhu,Ing,Andreas,Jensen,Modinos,Wang} and can be of the order of 0.4-0.8 eV (see e.g. Fig. 3 in Ref.\cite{Zhu}). This new degree of freedom...

4) It is not clear from the presented data whether there is a dependence of the current contrast on the applied voltage. It would also be more informative to plot some of the I(U) curves in semi-logarithmic coordinates so that the difference in current at low voltages is visible.

Our reply: in the new Figure (previously Fig.3, now labeled Fig.4) we have added the required semilogarithmic plots of the data as Fig.4c,d.

5) In Figure 11d, the current contrast appears to increase with Z. However, theory predicts the opposite relationship. It is worth commenting on this discrepancy.

Our reply. We have added the following sentence: “In these data, the contrast appears to increase with Z. However, we do not attach a significance to this apparent increase, as the uncertainty of the data is quite large. A further set of experiments which studies systematically the Z dependence is required to confirm or disprove the observed trend. What seems to be well established is that....”

6) It follows from Appendix D that the sign of the current contrast is determined by the sign of the difference between the work function, namely, the source current increases with a decrease in the work function of the collector. If this statement is true, then it should be noted in the main text.

Our reply. We have included the statement proposed above by the referee VERBATIM in the bulk of the paper!

7) I believe there is a mistake in Page 8 line 51 “we recall that $\phi_{\text{W}} - \phi_{\text{Fe/WC}}$ also has a negative sign”, however it has positive sign according to Page 8 line 43 “ $\phi_{\text{Fe/WC}} - \phi_{\text{W}} = -0.22\text{eV}$ ”. The same for “ $\phi_{\text{W}} - \phi_{\text{Au}}$ ”.

Our reply: The wrong sign was a typo. We have corrected it.